# Mind the truncation gap: challenges of learning on dynamic graphs with recurrent architectures.

**João Bravo**                                                      *joao.bravo@feedzai.com*
*Feedzai, Portugal*

**Jacopo Bono**                                                    *jacopo.bono@feedzai.com*
*Feedzai, Portugal*

**Pedro Saleiro**
*Feedzai, Portugal*

**Hugo Ferreira**                                                  *hugo.ferreira@feedzai.com*
*Feedzai, Portugal*

**Pedro Bizarro**                                                  *pedro.bizarro@feedzai.com*
*Feedzai, Portugal*

**Reviewed on OpenReview:** *https://openreview.net/forum?id=QezxDgd5hf*

## Abstract

Systems characterized by evolving interactions, prevalent in social, financial, and biological domains, are effectively modeled as continuous-time dynamic graphs (CTDGs). To manage the scale and complexity of these graph datasets, machine learning (ML) approaches have become essential. However, CTDGs pose challenges for ML because traditional static graph methods do not naturally account for event timings. Newer approaches, such as graph recurrent neural networks (GRNNs), are inherently time-aware and offer advantages over static methods for CTDGs. However, GRNNs face another issue: the short truncation of backpropagation-through-time (BPTT), whose impact has not been properly examined until now. In this work, we demonstrate that this truncation can limit the learning of dependencies beyond a single hop, resulting in reduced performance. Through experiments on a novel synthetic task and real-world datasets, we reveal a performance gap between full backpropagation-through-time (F-BPTT) and the truncated backpropagation-through-time (T-BPTT) commonly used to train GRNN models. We term this gap the "truncation gap" and argue that understanding and addressing it is essential as the importance of CTDGs grows, discussing potential future directions for research in this area.

## 1 Introduction

In many domains, data naturally takes the form of a sequence of interactions between various entities, which can be represented as an evolving network. Examples include interactions between users on social networks, card payments from users to merchants in payment networks, or bank transfers between entities in financial networks. These interactions typically involve only a pair of entities and are often modeled as a dynamic graph, where each entity is represented by a node and each interaction by an edge with an associated timestamp.

Unlike static graphs, where links between entities are persistent, dynamic graphs capture the sequential aspect of interactions, which can provide valuable information for different inference tasks. In these settings, the latent variables associated with nodes, such as user preferences in social networks, can evolve over time. Tasks on dynamic graphs involve predicting certain edge or node properties or the likelihood of an event involving two nodes at a specific point in time based on past information.

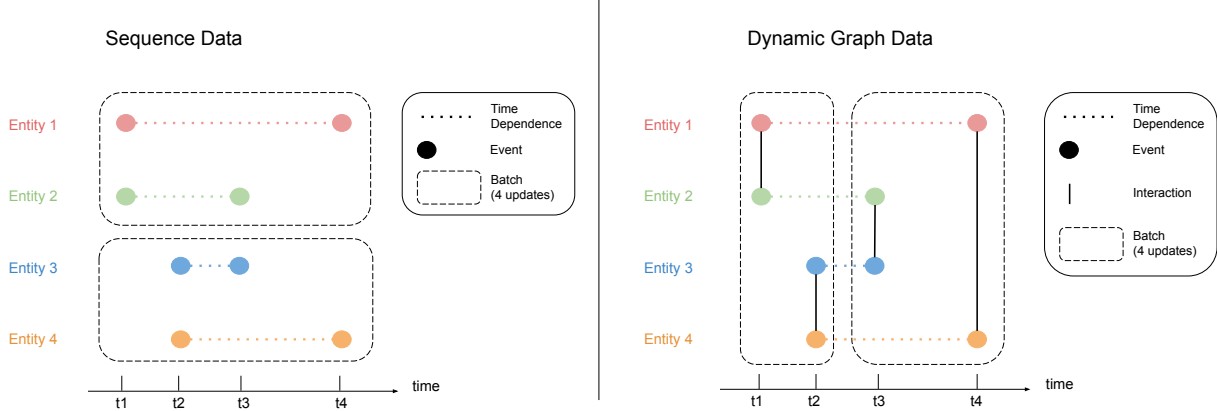

Figure 1: **Truncation of temporal history becomes severe in dynamic graphs.** (left) Sequence based data can be grouped by sequence when defining batches. In this specific example of sequences with two events, with a batch capacity of 4 entity updates, we can include two sequences per batch. Temporal dependencies between the events (horizontal dotted lines) are not broken by batching. (right) Due to the interactions between states, we cannot consider isolating a subset of entities in a batch on dynamic graphs as we need the counterparty entity's state to update an entity's state when an event occurs. Batches are defined by time instead of by entity but this leads to more extreme gradient truncation along the time axis (time dependencies are broken by batching). In the example, with the same capacity of 4 entity updates, each batch now includes only a single event per entity.

Notably, (Dai et al., 2017; Trivedi et al., 2019; Kumar et al., 2019; Rossi et al., 2020) have proposed treating such dynamic graphs as dynamical systems where a state for each entity evolves over time. The dynamics of these states are based on recurrent neural network (RNN) cells that use the previous states of both interacting nodes when computing the new states, therefore coupling both sides of an interaction at the time of an event. We call this class of models graph recurrent neural networks (GRNNs) because they are a straightforward extension of the RNNs used for sequence modeling to dynamic graphs.

Although there are differences in how various GRNN approaches handle training, all of them have in common that batches are formed using sequential time windows over the events. Backpropagation is applied to each batch to obtain the gradients with respect to the learnable parameters and therefore the learning signal cannot cross batch boundaries. In other words, this *truncation* of the backpropagation ignores the dependence of information older than the current batch.

In sequence modeling tasks, one typically has many sequences that are fully independent. If small enough, these sequences can be included in their entirety in a batch, or alternatively, they can be broken down into sizable chunks thus mitigating the impact of truncating the backpropagation (Figure 1, left). On dynamic graphs, however, it is not possible to split the data into different sequences per entity due to the coupled dynamics. Effectively, we have *a single global sequence* and batches need to be defined over globally ordered interaction events, from oldest to newest (Figure 1, right). Because such a sequential batch of events can involve completely different entities, if the number of entities is large enough one can find themselves in a situation where batches include a single update per entity. This in turn means that, from the entity point of view, only temporal dependencies spanning a single hop in the graph can be learned accurately. Such drastic truncation could lead to incorrect training gradients and a failure in learning to leverage longer term dependencies existing in the data.

The main goal of this work is to **investigate the impact of this truncation in GRNN models** by comparing the training approach proposed in previous works with F-BPTT. We propose a synthetic edge regression task on a dynamic graph that truncated backpropagation fails to learn when dependencies over

more hops are required. [1] We also show that a gap exists between the performance achievable with truncated and full backpropagation, on real world dynamic graph datasets by comparing both training methods on popular benchmarks. We call this drop in performance the *truncation gap*, since it is solely caused by the truncated backpropagation.

While F-BPTT is not a practical solution for large dynamic graph datasets due to its memory scaling, our results show that current training approaches based on truncating backpropagation are not able to fully exploit the capacity of GRNN models. We believe that further investigation into better training approaches for these models is warranted and propose possible future research directions.

## 2 Background

Initial approaches to dynamic graphs relied on static GNNs applied to different *snapshots* of the graph over chosen temporal windows. Because these are too dependent on the granularity of the time windows, ignoring the ordering of events within each window, other approaches that can more naturally cope with the sequential nature of edges were developed. These can be broadly categorized into transformer and RNN based approaches, mirroring the status quo in sequence modeling, together with a third class of methods based on random walks on the temporal graph.

Transformer based approaches such as TGAT (Xu et al., 2020), DyG2Vec (Alomrani et al., 2024) and DyGFormer (Yu et al., 2023) explicitly consider a temporal neighborhood as the context for each prediction, typically restricted to a few hops. While, by construction, this limits the extent of past information that the model can leverage, these methods don't suffer from the same issue as RNN based approaches where the computational graph that is back-propagated through is a truncated version of the computational graph used to arrive at a forward prediction. Furthermore, these approaches often still require some degree of approximation in the form of subsampling of the temporal neighborhood (also known as *neighborhood dropout*).

Random walk-based methods (Wang et al., 2022; Jin et al., 2022) take a similar approach where rather than sampling a neighborhood of a node of interest, temporal random walks over the dynamic graph are used to aggregate information that is relevant for inference. While a random walk offers a limited context for inference, because backpropagation is run on the same computational graph used during the forward pass, these methods don't suffer from the truncation issue that is the focus of this work.

### 2.1 Graph Recurrent Neural Networks (GRNNs)

RNN based approaches define a dynamical system over the network where the state comprises a hidden state for every node, $i \in [1..N]$, in the network: $\mathbf{h} = \left(\mathbf{h}^{(1)}, \ldots, \mathbf{h}^{(N)}\right) \in \mathbb{R}^{Nm}$. For a $k$-th event between a source node, $s_k$, and a destination node, $d_k$, at time $t_k$, the hidden states, $\mathbf{h}^{(s_k)}$ and $\mathbf{h}^{(d_k)}$ for the nodes involved in the interaction are updated using a pair of functions, $g_s$ and $g_d$:

$$
\begin{aligned}
\mathbf{h}_k^{(s_k)} &= g_s\left(\omega\left(\delta t_k^{(s_k)}\right), \mathbf{h}_{k-1}^{(s_k)}, \mathbf{h}_{k-1}^{(d_k)}, \mathbf{x}_k\right) \\
\mathbf{h}_k^{(d_k)} &= g_d\left(\omega\left(\delta t_k^{(d_k)}\right), \mathbf{h}_{k-1}^{(d_k)}, \mathbf{h}_{k-1}^{(s_k)}, \mathbf{x}_k\right)
\end{aligned}. \tag{1}
$$

Here, $\omega(\delta t)$ is an optional function producing an embedding for the time elapsed since the last update for each node, and $\mathbf{x}_k$ the message of the current interaction (which could itself be a combination of edge and node features). These functions are generally based on RNN cells such as LSTM (Hochreiter & Schmidhuber, 1997) or GRU (Cho et al., 2014), therefore, extending the use of RNNs to the dynamic graph setting. We will refer to these as GRNNs, noting that other names have been used in the literature, such as memory-based temporal graph neural networks (Zhou et al., 2023) and message passing temporal graph networks (Souza et al., 2022). A more general framework for GRNNs is discussed in Appendix A.

An early example of the GRNN approach is Wang et al. (2016) which proposes a temporal point-process model for event prediction where a latent state for each entity in the network is kept and updated on each

---

[1]The proposed synthetic task is available at `https://github.com/feedzai/truncation-gap`, as a benchmark of the ability of future approaches to learn temporal dependencies.

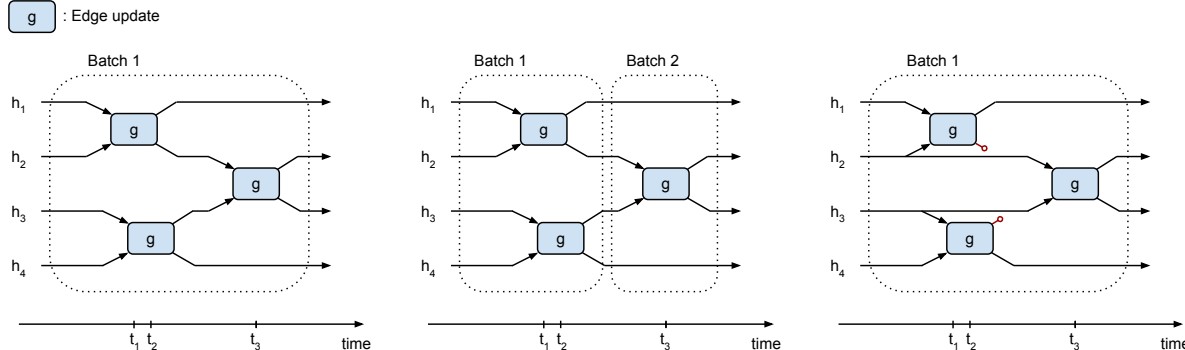

Figure 2: Three different batching strategies illustrated. Four nodes with respective states $h_1 \ldots h_4$ interact as in Figure 1 until time $t_3$. Each blue box denotes an interaction, where the hidden states of the two interacting nodes are updated. On the left the approach of Dai et al. (2017) where a different computation graph is built for each mini-batch. Due to the sequential processing within a batch, computational efficiency is lost. In the middle the *t-Batching* strategy of Kumar et al. (2019) which uses variable sized batches to guarantee no sequential dependencies within a batch, allowing parallel processing. On the right, the approach of Rossi et al. (2020) that uses fixed size batches and parallel processing at the cost of correctness, leading to inconsistent histories where the latent states used for the third event ignore the updates of the previous two events.

event with a coupled dynamic model based on a Hawkes process. Dai et al. (2017) extend this model with a more generic update based on a standard RNN cell and Trivedi et al. (2017) apply a similar model to dynamic knowledge graphs. JODIE (Kumar et al., 2019) uses a similar architecture but foregoes predicting the time of interaction, instead focusing on ranking how plausible interactions between pairs of entities are at a given time.

DyRep (Trivedi et al., 2019) and TGN (Rossi et al., 2020) constitute hybrid solutions, incorporating transformer elements into a GRNN architecture. DyRep proposes a recurrent model where the messages coming from the counterparty are itself the result of a graph attention operation over the counterparty graph neighborhood. TGN introduces instead an attention based *embedding module* on top of the recurrent *memory* model to deal with the staleness problem of the hidden states in the absence of events.

Unlike previous approaches which require access to the temporal graph at inference time and perform complex operations over it, for pure GRNNs the graph itself does not need to be explicitly stored. Instead, it suffices to store the node embeddings and update them using the recurrent architecture whenever a new edge arrives. This makes these models appealing for inference scenarios requiring low-latency such as fraud detection.

Another advantage of GRNNs over transformer or random walk models is that they would seem to promise an infinite causal context towards the past. However, due to the way they are trained, the ability to leverage information that is far in the past can be compromised as we discuss in the next subsection.

## 2.2   Training GRNNs: Batch Processing Strategies

RNNs are typically trained with Back Propagation Through Time (BPTT) or a truncated version of this algorithm, T-BPTT, when sequences become too long. In dynamic graphs, however, there is no good option to define sequences, since nodes interact with each other (Figure 1). Because future events are influenced by past events, the natural way to define batches is by adopting a sliding window over the globally ordered events. In other words, no random shuffling of events or sequences is possible, and each epoch will contain batches of sequential events between any pair of nodes in the graph. There are different approaches in the literature on how to define and process these batches:

**Deep CoEvolve and DyRep.** Dai et al. (2017) and Trivedi et al. (2019) consider sequential batches of interactions. A computation graph for the forward pass is built for each batch, processing each event in the batch in sequence (Figure 2.2, left). This graph is then back-propagated through. Gradient propagation is therefore restricted to each batch, ignoring the model parameter dependencies of the hidden states of each entity that serve as inputs to this computational graph. When batches are large enough, this sequential processing of each event in a batch can allow for the learning of multi-hop temporal dependencies, but at the cost of being very slow (essentially each edge in the dataset is processed sequentially). Furthermore, for large networks and if events are well distributed across many entities, a single update per entity would be processed in each batch and therefore only a single hop temporal dependency is learned.

**JODIE.** Kumar et al. (2019) propose instead using variable sized batches so that a single update per entity is present in each batch (which they call *t-Batches)*, Figure 2.2, middle). This makes the training more efficient since all updates in a single batch can now occur in parallel, but also exacerbates the problem of gradient truncation because it guarantees a truncation horizon of a single hop per entity.

**TGN.** Rossi et al. (2020) consider fixed size batches of temporally ordered events, but ignore any sequential dependencies within each batch by processing all updates in parallel (Figure 2.2, right). When multiple events involving a single entity are present in a batch, this leads to an inconsistent history where the last edge effectively determines the final state of the node, ignoring previous edges that occurred in the same batch. While this approach makes the processing more efficient and straightforward, it (1) guarantees a truncation horizon of a single hop and (2) introduces an approximation that can potentially deteriorate the results. This forces the authors to propose keeping batch sizes large enough to be more efficient than (Deep)CoEvolve (Dai et al., 2017), but small enough to avoid serious deterioration of results. Empirically, they found that a batch size of 200 provided a reasonable trade-off. We also note that the GNN module on top of the recurrent cell may help to alleviate the deterioration, since it provides an alternative way to model dependencies in the data to the GRNN component.

In summary, *all discussed methods suffer from truncating backpropagation with typically small horizons.* Because this flaw is tightly linked to the use of backpropagation, and due to the success of backpropagation in other settings and the ease of implementation in current deep learning frameworks, it has been accepted as an unavoidable shortcoming when using GRNNs. In fact, none of the above works discusses this truncation's effect on learning. In the next section, we propose a synthetic task to benchmark performance of the various approaches on learning temporal dependencies. We identify severe degradation of performance due to the truncation, and subsequently discuss future research directions to mitigate this issue.

## 3 Synthetic Task

In order to investigate the effect of limiting the backpropagation to a single batch in GRNNs on their ability to learn longer term dependencies we propose a novel edge regression synthetic task. We construct the task to achieve the following desirable properties:

- A parameter $M$ should regulate the amount of past edge memory necessary to solve the task (larger $M$ would denote older information is needed to correctly predict the label).

- The latent state of the source node is updated using the destination node's latent state and vice-versa. Otherwise, non-graph based methods would be able to solve the task.

The code to generate the data will be made available upon acceptance, such that this task can be used in future research to benchmark the developed methods.

### 3.1 Task Specification

We took inspiration from the *adding task* (Hochreiter & Schmidhuber, 1997), a benchmark frequently used for recurrent neural networks, and construct a variation for graphs that satisfies the above properties. We

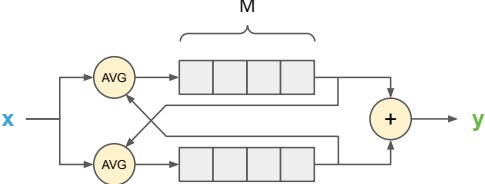

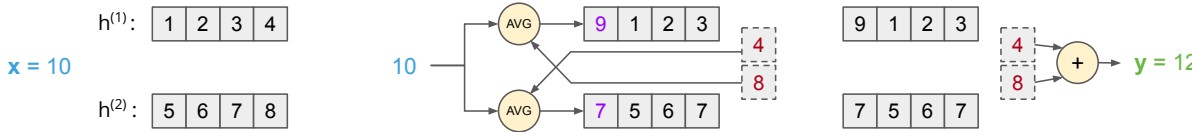

a) New interaction between two nodes    b) Average with **x** to update internal states    c) Compute output

Figure 3: Visual depiction of synthetic task dynamics. A fixed size FIFO buffer with length M (4 in the picture) is used to store the internal state of each node. When a new edge between two nodes occurs (a), the input is averaged with the last elements of each counterparty node's buffer in order to determine the new number to be stored in each buffer (b). The sum of these last elements also determines the output for the edge (c).

consider an internal state for each node $i \in [1..N]$ in a graph, consisting of a memory buffer of size $M$, $\mathbf{h}^{(i)} \in \mathbb{R}^M$. Each edge, $e_k = (s_k, d_k, x_k, y_k)$, is defined by two nodes, $s_k$ and $d_k$, an input numeric feature $x_k$ and a target $y_k$. The target is computed by adding the last two elements of the buffers of the nodes (prior to the state update for event $k$):

$$y_k = \left(\mathbf{h}_{k-1}^{(s_k)}\right)_M + \left(\mathbf{h}_{k-1}^{(d_k)}\right)_M .$$

Here, $\left(\mathbf{h}_{k-1}^{(s_k)}\right)_M$ denotes the $M$th element of the internal state for node $s_k$ (superscript) at step $k-1$ (subscript). The memory buffer of each node is updated after each edge arrival by shifting the values in the buffer using a first-in-first-out (FIFO) principle. Importantly, the newly stored value depends on the edge feature and the *counterparty* node, such that the second property above is satisfied:

$$\left(\mathbf{h}_k^{(s_k)}\right)_0 = \frac{1}{2}\left(\mathbf{h}_{k-1}^{(d_k)}\right)_M + \frac{1}{2}x_k$$
$$\left(\mathbf{h}_k^{(s_k)}\right)_i = \left(\mathbf{h}_{k-1}^{(s_k)}\right)_{i-1}, \quad i \in [1\ldots M] .$$

The update is symmetrical for $\mathbf{h}^{(d_k)}$. A schematic representation of the task is depicted in Figure 3.1.

This task becomes more challenging as $M$ increases since it determines a delay (in number of edges) between an input being observed and it affecting an output for the same node. A recurrent model will have to learn the underlying dynamics of the task from these input/output pairs in order to correctly solve it. It will, therefore, have to memorize older information for each node as $M$ increases.

To generate dynamic graphs for this synthetic task we sample edges randomly by picking a random pair of nodes, $s_k$ and $d_k$, uniformly and a corresponding random edge feature, $x_k \sim_{i.i.d.} \mathcal{N}(0,1)$, drawn from a standard normal distribution. The states for all nodes are initialized to the zero vector. For our experiments we consider dynamic graphs consisting of 1000 edges and using a total of 100 nodes, which we refer to as one epoch. We generate a new dynamic graph per epoch.

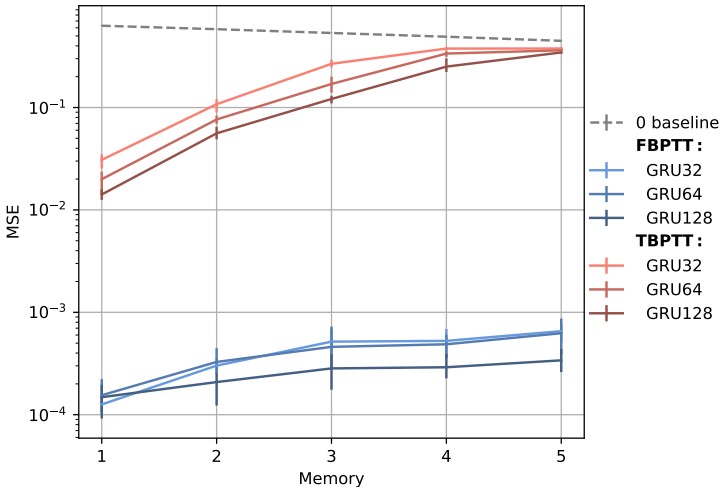

Figure 4: Mean squared error (MSE) obtained for different sized GRU models trained with both F-BPTT and T-BPTT. The depicted error bars correspond to the min and max MSE values over 5 different random seeds used for parameter initialization and dynamic graph sampling. The solid lines correspond to the mean MSE over the different seeds.

### 3.2 Results

All models are based on Equation 1 using a single GRU cell for both updates (since the dynamics are symmetric) and without time encoding.

We train a first model with F-BPTT by back-propagating through an entire epoch (i.e. the entire dynamic graph) in order to compute the gradient of the total loss (sum of losses for all edges). One gradient step is taken per epoch using an AdamW optimizer Loshchilov & Hutter (2019) with a learning rate of 1e-3 and weight decay of 1e-4 for a total of 5000 epochs.

For the model using T-BPTT we compute truncated gradients for each edge and accumulate these (i.e., sum them) over an entire epoch, performing a single gradient step at the end. We use the same optimizer and hyperparameters for the same number of epochs as with F-BPTT. The reasoning behind this setup is to be directly comparable with F-BPTT but, in this instance, employing an online setup where a gradient step is taken per edge did not seem to significantly alter the results for T-BPTT.

We train models using F-BPTT and T-BPTT and with hidden state sizes of 32, 64 and 128. For all models, we compute the average loss over the last 100 epochs across various values of the memory parameter $M$. For comparison, we also plot the performance of a trivial baseline that predicts $\hat{y}_k = 0$ for every edge.

While for $M = 1$ the models using T-BPTT are able to approximately solve the task, their performance rapidly degrades as $M$ is increased. In contrast, models trained using F-BPTT are able to fully solve the task (the final loss is orders of magnitude smaller than the target variance) for every value of $M$ tested (Figure 3.2).

## 4 Dynamic Graph Benchmark Results

In order to show that the *truncation gap* is also present in real world dynamic graphs we conduct a simple experiment on three publicly available dynamic graph datasets from Kumar et al. (2019) (Reddit, Wikipedia and MOOC).

Table 1: Search space for the 25 trials of random search used to tune the hyperparameters of the optimizer and the model dropouts. The last parameter refers to a uniform at random choice between the two types of state dropout.

| Parameter | Domain | Distribution |
|---|---|---|
| learning_rate | $[10^{-3}, 10^{-2}]$ | Log-Uniform |
| weight_decay | $[10^{-5}, 1]$ | Log-Uniform |
| mlp_dropout | $[0, 0.3]$ | Uniform |
| state_dropout | $[0, 0.3]$ | Uniform |
| state_dropout_type | {Regular, Recurrent} | Uniform |

We consider a GRNN model based on a GRU cell with a hidden state size of 64. Similar to Rossi et al. (2020) we use the same GRU for the updates to both source and destination nodes (note that despite these nodes being heterogeneous, using a GRNN model with non-symmetric dynamics did not lead to better performance). We also disregard any timestamp information on the edges by not having our update function depend on any $\delta t$, therefore assuming that there are no autonomous dynamics between events. These choices are made to simplify the experimental setup since our goal is not to challenge the state of the art results for dynamic graphs. Our choice of state size is limited by the memory required to run full backpropagation on the training set.

We train the GRNN model with both truncated and full BPTT, using the same batching strategy of Rossi et al. (2020) (see Section 2.2) with a batch size of 200. We note that while this strategy introduces some error by potentially ignoring some updates in each batch, it is the most efficient and practical to implement, allowing us to run full BPTT with a reasonable hidden state size on these benchmark datasets. We also employ the same negative sampling strategy where for every edge in the graph, a negative edge is obtained by uniformly sampling a random destination node. The training loss is then the binary cross-entropy for classifying positive and negative edges. For F-BPTT we backpropagate through all the batches in the training data, computing the gradient of the total training loss for the epoch. This is possible for the selected datasets due to their relatively small size. In order to enable a fair comparison, we also accumulate the truncated gradients for every batch computed with T-BPTT, performing a single parameter update per training epoch. This allows us to have a common training setup with the same number of epochs and the same number of parameter updates for both methods with the same potential choices of hyperparameters.

We use the Adamw optimizer (Loshchilov & Hutter, 2019), and tune the learning rate and weight decay parameters using 25 trials of random search. We also tune the values for the dropout of the multilayer perceptron classifier as well as a state dropout that is applied to the node states modified in each batch. For this state dropout we experiment with two approaches: (i) a regular dropout layer applied directly to all the states updated at each batch or (ii) a layer that keeps each element of the previous state with a given dropout probability instead of updating to the newly computed state (which we call recurrent dropout). The search space used for these tuned parameters is summarized in Table 1. We repeat our experiments using 3 different seeds determining the hyperparameter sampling, model initialization, dropout and negative edge selection during training for the Wikipedia and MOOC datasets. A single seed is used for Reddit, which is the largest dataset, due to it being substantially slower to run.

For evaluation we use a similar setup to Kumar et al. (2019) and Huang et al. (2023). We process the validation and test sets using the same batching strategy used during training, and then rank for every edge all the possible destination nodes according to the probabilities predicted by the model. Note that since the number of destination nodes for these datasets is at most 1000 we don't use any negative sampling for evaluation. Mean Reciprocal Rank (MRR) and Recall@10 values are computed based on the rank of the true destination node for each edge. We stop each trial when neither metric has improved for 250 epochs.

The results are reported in Table 2, where we can observe a large *truncation gap* on the Reddit and MOOC datasets representing at minimum a 10% improvement when using F-BPTT. For Wikipedia, while a 3% improvement in the Recall metric was also observed, a non-statistically significant (negative) result is obtained for MRR.

Table 2: **Truncated BPTT results in worse performance across most datasets and metrics.** Results on the benchmark dynamic graph datasets showing that there is a performance gap between the models trained with truncated and full backpropagation. For Wikipedia and MOOC, the reported values are the means and standard errors on the test set over the different seeds.

| | Reddit | | Wikipedia | | MOOC | |
|---|---|---|---|---|---|---|
| | MRR | Recall@10 | MRR | Recall@10 | MRR | Recall@10 |
| T-BPTT | 0.549 | 0.717 | $0.534_{\pm 0.011}$ | $0.677_{\pm 0.007}$ | $0.281_{\pm 0.006}$ | $0.643_{\pm 0.021}$ |
| F-BPTT | 0.648 | 0.803 | $0.525_{\pm 0.012}$ | $0.698_{\pm 0.006}$ | $0.343_{\pm 0.008}$ | $0.708_{\pm 0.007}$ |
| Truncation Gap | 0.010 | 0.085 | $-0.009_{\pm 0.012}$ | $0.022_{\pm 0.006}$ | $0.062_{\pm 0.006}$ | $0.064_{\pm 0.020}$ |
| Improvement (%) | 18.1 | 11.9 | $-1.8_{\pm 2.2}$ | $3.2_{\pm 0.8}$ | $22.2_{\pm 2.6}$ | $10.1_{\pm 3.5}$ |

## 5 Beyond Backpropagation

While we argued that recurrent neural network architectures are well-suited to handle tasks on dynamic graph data, we have shown that the truncation present in state-of-the-art approaches can lead to the failure of learning tasks requiring dependencies more than a hop away. Since the truncation is inevitable in approaches using backpropagation, solving the issue warrants looking beyond backpropagation-based methods.

Viable research directions could include approximations to real-time recurrent learning (RTRL). More specifically, Tallec & Ollivier (2017); Mujika et al. (2018); Benzing et al. (2019) describe unbiased stochastic low-rank approximations to the true Jacobians in recurrent neural networks on sequential data, which make online learning feasible. Adapting such methods for the dynamic graph settings would be an interesting avenue to resolve the temporal dependency learning issue. Another approach would be to adopt simpler recurrent architectures that make RTRL feasible, similar to what is proposed in Zucchet et al. (2023) in the context of sequence modeling.

## 6 Conclusions

We surveyed current methods leveraging recurrent architectures for inference tasks on dynamic graphs, discussing the strengths and weaknesses of various training and batching strategies. We identified that backpropagation may reach its limits in this dynamic graph setting, due to a severe truncation of the history, confirming that it is holding back GRNN models both in a proposed novel synthetic task and in publicly available real-world datasets. Therefore, we believe that methods beyond backpropagation warrant more attention in this context.

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

# A  A General Framework for GRNNs

One can define a dynamical system over the graph with a hidden state comprising a hidden state for every node, $i \in [1 \ldots N]$, in the network: $\mathbf{h}(t) = \left(\mathbf{h}^{(1)}(t), \ldots, \mathbf{h}^{(N)}(t)\right)$. Between edges (i.e. interaction events), this state can be subject to an autonomous evolution which we will assume to be time-invariant and similar and independent for each node in the network:

$$\mathbf{h}^{(i)}(t) = f\left(t - \tau, \mathbf{h}^{(i)}(\tau)\right). \tag{2}$$

We can visualize this evolution as the blue trajectories in Figure 5. Here we express this time evolution in integral form, but it could instead be given as an ordinary differential equation such as a neural ordinary differential equation (NeuralODE) (Chen et al., 2019). Note that if the network is heterogeneous, (i.e., there are different types of entities), we could have different evolution laws for each entity type.

An edge, $e_k = (s_k, d_k, t_k, \mathbf{x}_k)$, is defined by a source entity, $s_k \in \mathcal{H}$, a destination entity, $d_k \in \mathcal{X}$, a timestamp, $t_k \in \mathbb{R}$ and a set of input features, $\mathbf{x}_k \in \mathcal{X}$. The states for both nodes associated with the edge are updated by a function, $g : \mathcal{H} \times \mathcal{H} \times \mathcal{X} \rightarrow \mathcal{H} \times \mathcal{H}$. This update allows information encoded in the state of one node to be propagated to the counter-party node of the edge and vice-versa. We assume this update to be time invariant for simplicity:

$$\left(\mathbf{h}^{(s_k)}(t_k^+), \mathbf{h}^{(d_k)}(t_k^+)\right) = g\left(\mathbf{h}^{(s_k)}(t_k^-), \mathbf{h}^{(d_k)}(t_k^-), \mathbf{x}_k\right) \tag{3}$$

$$\mathbf{h}^{(i)}(t_k^+) = \mathbf{h}^{(i)}(t_k^-), \quad i \notin \{s_k, d_k\}. \tag{4}$$

We can visualize this update as the red arrows in Figure 5.

In most settings, the state of a node is only needed for the end task at a time of an interaction involving that node. In this case, we can simplify the above model by merging the autonomous evolution $f$ together with the state update $g$ into a single update function $h$. There are two possibilities for this, depending on whether one needs the state immediately before or after a potential interaction for the end-task. As an example, for an edge classification or regression task where the target for the edge may depend directly on the edge

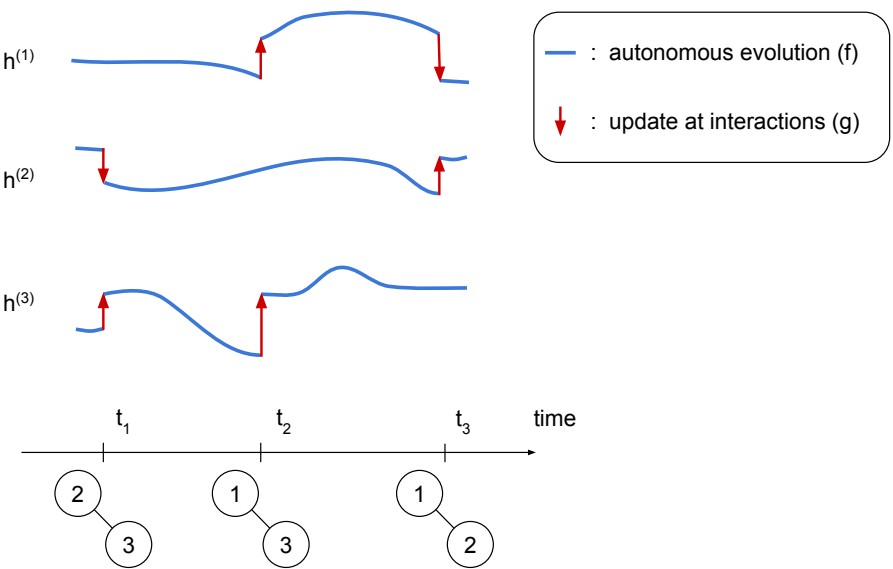

Figure 5: The dynamical system over the hidden states can in general contain two components: a function $f$ encoding the evolution between interactions (blue), and a function $g$ encoding the updates due to interaction events (red). In this example interactions happen at times t1, t2 and t3 between nodes 2-3, 1-3 and 2-3 respectively.

features we could use the node states after the update. We can denote by $\tau^{(i)}(t)$ the last timestamp before $t$ where an edge involving node $i$ was observed and $\delta t_k^{(i)} = t_k - \tau^{(i)}(t_k)$ and write:

$$\left(\mathbf{h}_k^{(s_k)}, \mathbf{h}_k^{(d_k)}\right) = h\left(\delta t_k^{(s_k)}, \delta t_k^{(d_k)}, \mathbf{h}_{k-1}^{(s_k)}, \mathbf{h}_{k-1}^{(d_k)}, \mathbf{x}_k\right) , \tag{5}$$

where we also defined $\mathbf{h}_k^{(s_k)} = \mathbf{h}^{(s_k)}\left(t_k^{(s_k)+}\right)$. Here $h$ is given by applying first $f$ starting from the state immediately after the previous update of each node and then $g$ for the edge update. This is depicted on the left (a) of Figure 6 and can be written as

$$h\left(\delta t_k^{(s_k)}, \delta t_k^{(d_k)}, \mathbf{h}_{k-1}^{(s_k)}, \mathbf{h}_{k-1}^{(d_k)}, \mathbf{x}_k\right) = g\left(f\left(\delta t_k^{(s_k)}, \mathbf{h}_{k-1}^{(s_k)}\right), f\left(\delta t_k^{(d_k)}, \mathbf{h}_{k-1}^{(d_k)}\right), \mathbf{x}_k\right) . \tag{6}$$

For a link prediction scenario one would instead like to answer the question of whether a link between two nodes is likely at a given time (or simply rank how likely potential links between different nodes are). For this task, the node states immediately before a (potential) interaction are necessary since no information about the (potential) edge can be considered. We can thus redefine $\mathbf{h}_k^{(s_k)} = \mathbf{h}^{(s_k)}\left(t_k^{(s_k)-}\right)$ and $\delta t_k^{(i)}$ to be the time between edge $k$ and a potential future interaction in Equation 5. $h$ is then given by first applying the node updates, $g$, and then propagating the states using $f$. This is depicted on the right (b) of Figure 6. Note that in a practical application, the update for event $k$ for a node $i$ would only be computed at the time of a future potential event for node $i$, thus requiring that extra memory of the last event for each node be maintained. In the example of Figure 6, the update for event 1 could be computed at time $t_3$ for node 2 and $t_2$ for node 3. This is the approach taken in Rossi et al. (2020) where the update is potentially recomputed several times with different values of $\delta t$ corresponding to the times that node $i$ is evaluated as a possible link for another node (i.e., it is selected as a negative example) and when a future edge involving node $i$ occurs.

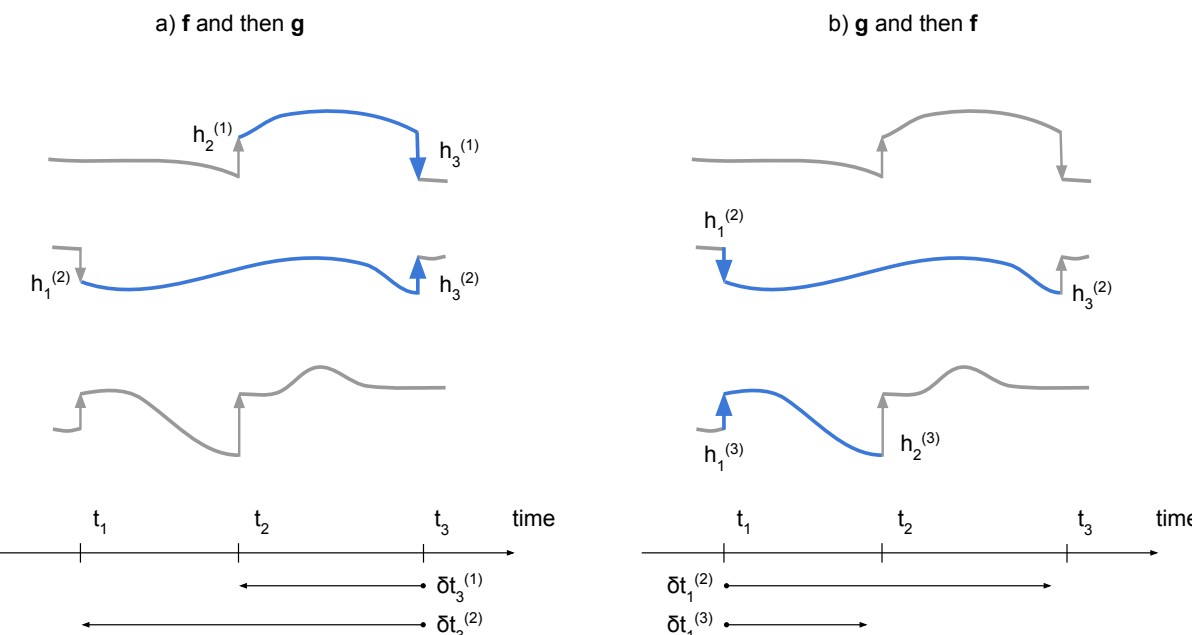

Figure 6: The two different possibilities for merging the autonomous evolution and event update into a single function. On the left (a) in blue is the update corresponding to the third event between node 1 and 2. The state of each node immediately after its last update is first propagated using the autonomous evolution, $f$, up to $t_3$ and then the update, $g$, is computed. On the right (b) in blue is the update corresponding to the first event between nodes 2 and 3. The state of each node immediately before the event is first updated using $g$ and then propagated using $f$ up until the next (potential) event where the state is needed. Note the different definitions of $h_k^{(i)}$ and $\delta t_k^{(i)}$. On the left they refer to the state immediately after event $k$ and the time elapsed since the last update of the node before event $k$ respectively. On the right they refer to the state immediately before event $k$ and the time it will take until the next (potential) update of each node after event $k$ respectively.

