# OpenReview forum: "Mind the truncation gap: challenges of learning on dynamic graphs with recurrent architectures"
_TMLR — Accepted by TMLR_

### Review · Reviewer_Y1ML · 2024-10-09

**Summary Of Contributions:**

The paper identifies and addresses the "truncation gap" in training Graph Recurrent Neural Networks for dynamic graphs, demonstrating the impact of backpropagation truncation on learning temporal dependencies and proposing future research directions to mitigate this issue.

**Audience:**

Yes

**Claims And Evidence:**

Yes

**Requested Changes:**

See above.

**Strengths And Weaknesses:**

Strengths:
- The paper addresses a critical but often overlooked issue in the training of Graph Recurrent Neural Networks (GRNNs) for dynamic graphs, specifically the impact of truncated backpropagation-through-time (BPTT).
- The paper provides empirical evidence from both synthetic and real-world datasets, demonstrating the practical implications of the truncation gap on model performance.

Weaknesses:
-  While the paper identifies the truncation gap, it could provide a more detailed explanation of how exactly truncated BPTT impacts the learning of temporal dependencies in GRNNs. Can the authors elaborate on the mechanics of this impact?
- The paper notes the use of a simple model for the experiments. How would more complex models, capable of capturing deeper dependencies, fare with truncated BPTT? Does the truncation gap increase with model complexity?
- While the paper focuses on the truncation issue, it does not delve deeply into the computational complexity of GRNNs, particularly how this complexity scales with the size of the dynamic graph.
- The paper focuses on RNN-based methods. How do the findings compare to non-recurrent methods that may be used for dynamic graph data? Is the truncation gap a problem unique to RNNs, or is it a more general issue?
- The paper could benefit from a comparison with other models that do not use BPTT to understand how unique the identified truncation gap is to GRNNs.

---

> ### Author Response · Authors · 2024-10-31
> **Response to Reviewer**
>
> We thank the reviewer for their time and address their comments below.
>
> **On the mechanics of the truncation gap issue**
>
> The mechanics of this truncation are the same as in RNNs [(Tallec & Ollivier, 2017)](https://arxiv.org/abs/1705.08209). The issue lies with the fact that the computational graph that is used in the forward pass to arrive at a prediction is truncated in the backwards pass, leading to an incorrect computation of gradients. Namely, at some point in the sequence going backwards, the hidden state is assumed to be a constant independent of the model parameters and backpropagation terminated. This is often a necessity when dealing with long sequences but can bias the computed gradients to ignore directions in parameter space that can have longer term impact.
>
> In the case of dynamic graphs this issue is significantly more serious since batches often contain a single update per node as opposed to sequence modeling where truncation lengths are often in the thousands of elements per sequence.
> The issue of truncated back-propagation-through-time in learning temporal dependencies is now discussed in paragraph 4 of the introduction.
>
>
> **On comparing to non-GRNN models**
>
> The truncation gap problem as discussed in our work is exclusive to recurrent architectures on graphs.
> Our intention was merely to shed light on this issue that affects one of the most popular classes of models for dynamic graphs, showing that it can have a real impact, not only on a constructed scenario but also in a real world setting.
>
> More concretely, we have adapted our work to reflect this message:
> - Adapting the title of the paper.
> - Stating the aim clearly in the introduction.
> - Discussed this more at length in the background section.
>
> **On the impact of model size on the truncation gap**
>
> In our toy data experiment we try different model sizes and find that it has little influence on the truncation gap.
> Unfortunately, for our experiments with real data we are limited in how large we can make the model while still being able to do full backpropagation. However, we note again (see answer to the first point), that regardless of the model size, backpropagation is restricted to a batch and therefore can never cross a batch boundary. If a target depends on information outside of a batch, one is unlikely to learn dynamics that leverage that information using truncated BPTT, whether the model has more capacity or not.

---

### Review · Reviewer_A1Qi · 2024-10-17

**Summary Of Contributions:**

This work revisits the truncation gap problem for batched training of RNN-based Temporal GNNs. The authors review eight related works and summarize their architecture designs and batch-processing strategies. Experiments on synthetic and real-world tasks are performed where results of BPTT for one event and BPTT for the whole graph are compared.

**Audience:**

Yes

**Broader Impact Concerns:**

I do not have concerns on the ethical implications of this work.

**Claims And Evidence:**

No

**Requested Changes:**

The truncation gap problem in Temporal GNNs is a good research direction, I suggest the authors start from scratch and submit to another conference/journal.

**Strengths And Weaknesses:**

Strengths:
1. This manuscript is easy to follow.
1. The figures are clear and intuitive, conveying key concepts effectively.

Weaknesses:
1. The truncation gap of RNNs is a well-studied problem. Relevant works even date back to the 90s [1-3]. The major issues of BPTT with short truncation length are (1) the limited temporal context, which makes it hard to capture long-term dependencies, (2) vanishing/exploding gradients, and (3) overfitting to short-term patterns. However, none of them are discussed in this work.
1. The related work simply lists eight past works and introduces their architecture designs and batch-processing strategies one by one. If the survey is one of the major contributions, readers are expected to see a taxonomy study and an in-depth discussion of the key design choices supported by experiment results.
1. Important related works [4, 5] in the Temporal GNN domain that do not use RNN to capture temporal relationships are missing. The advantages and disadvantages of those works need to be discussed in detail and supported by experiment results.
1. Evaluation only includes T-BPTT and F-BPTT on limited datasets, which is incomplete. In addition, there is no analysis of the amount of temporal dependencies and their durations in each dataset. Why is there no improvement on the Wikipedia dataset? Is there anything unique about this dataset? The readers expect an evaluation that covers a granular grid in the dimensions of different datasets, different network architectures, different batching strategies, and different truncation lengths, with a few hypotheses that shed light on future Temporal GNN design. Accuracy, runtime, and convergence speed should all be compared. Even if simply altering the `--backprop_every` parameter in the TGN [code](https://github.com/twitter-research/tgn/blob/master/train_self_supervised.py#L36C24-L36C38) might provide some insightful findings, which would not cost more than a few hours to run.

[1] Learning long-term dependencies with gradient descent is difficult, https://ieeexplore.ieee.org/document/279181

[2] Memory Augmented Neural Networks with Wormhole Connections, https://arxiv.org/pdf/1701.08718

[3] An Empirical Exploration of Recurrent Network Architectures, https://arxiv.org/pdf/1701.08718

[4] Inductive Representation Learning on Temporal Graphs, https://arxiv.org/abs/2002.07962

[5] Do We Really Need Complicated Model Architectures For Temporal Networks?, https://arxiv.org/abs/2302.11636

---

> ### Author Response · Authors · 2024-10-31
> **Response to Reviewer**
>
> We thank the reviewer for their thoughtful review of our work. We address the reviewer's comments below.
>
> **Motivation**
>
> We agree with the reviewer that truncation of back-propagation through time is a known issue in RNNs (e.g., [(Tallec & Ollivier, 2017)](https://arxiv.org/abs/1705.08209)).
> Our main point was to highlight that popular architectures for dynamic graphs, such as JODIE [(Kumar et al., 2019)](https://arxiv.org/abs/1908.01207) and TGN [(Rossi et al., 2020)](https://arxiv.org/abs/2006.10637), truncate temporal dependencies to a single batch.
> Since a batch often includes a single update step per entity (node), this would be analogous to truncating BPTT to a single step in an RNN setting which could be considered quite extreme.
> Despite this, no work proposing recurrent architectures for dynamic graphs has ever discussed the potential impact of such truncation.
>
> Because these architectures have claimed SoTA results in the past, and are still very relevant today, we believe our work is important in shedding light on this issue.
> Namely, we show that truncation can indeed limit the performance achievable by these models in practice, thus suggesting a possible direction for improvement. We believe this is now more clear in the introduction.
>
> **Not Covering non-GRNN Approaches**
>
> As the reviewer points out, the truncation gap, as we define it, only applies to a specific type of recurrent architecture which we call graph recurrent neural network (GRNN).
> Transformer and attention-based architectures are explicit about using a finite context window (i.e., a finite neighbourhood of each node at any point in time) and therefore don't suffer from incorrect gradient computations from truncating back-propagation.
> In contrast, GRNNs would seem to promise an infinite context window but, in practice, training with truncated back-propagation can impact the learning of long-term dependencies.
> Our intention was not to evaluate or discuss the relative merits between these different approaches to dynamic graphs but merely to identify a problem with a particular type of approach (GRNNs). This is why we originally limited our survey to methods that can be classified under this umbrella. We have however expanded our background section to acknowledge other types of architectures and some of their pros and cons when compared with recurrent architectures.
>
> More concretely, we have adapted our work to better reflect this message:
> - Adapting the title of the paper.
> - Stating the aim clearly in the introduction.
> - A more in-depth discussion in the background section.
>
> **Limited experimental setup**
>
> The goal of our experimental setup is not to chase SoTA results on dynamic graphs but merely to highlight the issues with truncation.
> We do so by proposing a toy task that requires learning long-term dependencies in the graph.
> We also show that truncation can in fact have an impact on the performance achieved by GRNN models in real world datasets by comparing against full back-propagation.
> However, as we note in the paper, full back-propagation is not a viable solution as it is not computationally feasible on large graphs. This is the reason we restrict ourselves to these smaller benchmark datasets.
>
> Regarding the results on the Wikipedia dataset, we note that there is still a gap when considering the recall@10 metric even if it is smaller than other datasets. Furthermore, full backpropagation is able to achieve a considerably lower training loss but unfortunately this does not translate to a significant gap in test performance indicating that it is easier to overfit in this dataset.
>
> We also want to stress that on these benchmark datasets, we do not know the ground-truth in terms of temporal dependencies. It is plausible that on Wikipedia, longer temporal information is not as relevant for the link prediction task, which in turn would result in a small or no truncation gap. Our synthetic task was designed to isolate that dependence on required context more clearly, while we believe the benchmark datasets allow us to show that even in real-world scenarios (but not necessarily all) a gap exists.

---

### Review · Reviewer_i7NF · 2024-10-17

**Summary Of Contributions:**

The summary of the contributions of the paper is as follows:

- The paper targets to study the Continuous-time dynamic graphs (CTDGs) which are increasingly important for modeling complex systems in various domains where interactions evolve over time.

- It is indicated that the static graph methods struggle to accurately represent these dynamic interactions because they fail to incorporate the timing of events, which is crucial for understanding the dynamic behavior of domains represented by CTDGs.

- Graph recurrent neural networks (GRNNs) have emerged as a promising approach for modeling CTDGs, as they are specifically designed to be time-aware, allowing them to better capture the correlations of evolving interactions.

- It is highlighted that despite the advantages, GRNNs face a significant challenge due to the short truncation of backpropagation-through-time (BPTT). The paper claims the short truncation can hinder the ability of the models to learn and retain important temporal dependencies present in the data to a significant extend resulting in reduced performance. It is noted that the paper is the first to investigate the shortcoming, and present analysis on the truncation gap.

- Through emprical invesitagtions, the authors support the claim through revealing a notable performance gap—referred to as the "truncation gap"—between the results obtained using full backpropagation-through-time (F-BPTT) and those achieved with truncated backpropagation-through-time (T-BPTT) using a synthetic task and , highlighting the limitations of the latter.

- The authors believes that as the relevance of CTDGs continues to grow in various fields, it becomes crucial to understand and address this truncation gap to enhance the performance and applicability of machine learning models in dynamic graph scenarios.

- The paper concludes by discussing potential research avenues aimed at mitigating the truncation gap, suggesting that further exploration in this area could lead to more effective methods for handling temporal dependencies in CTDGs.

**Audience:**

Yes

**Claims And Evidence:**

No

**Requested Changes:**

Based on the aforementioned strenghts and weaknesses, the requested changes are provided in a orderly manner from the highest to the lowest importance as follows:
- The overall claim of the paper needs to be lowered and limited to memory-based approaches through a RNN point-of-view. This needs to be clearly communicated that this study does not cover non-memory-based approaches where some of the assumptions are not valid, for instance, sequntial/chronological training, caching of hidden states, etc. If it is decided to cover boarder categories of CTDG models, the theoretical aspect and the formulation behind the fundamentals of learning then would need to be improved to cover non-RNN approaches (e.g. Transformer-based). It is not clear if the truncation gap is only limited to memory-based appraoches, if Transformer-based approaches suffers from it as well, etc. Since memory-based appraoches are inefficient due to the two shortcomings, if Transformer-based appraoches does not have truncation gap to a significant extedn, then the motivation is voided and the claim of the study is invalid.
- The supporting evidence needs to be improved to be able to support the claim in the following directions:
  1. aligning the experimental setup with the one used in the literature, for instance consider temporal information of the interactions during the encoding.


  2. Show the truncation gap on the real dataset for the baseline having the same performance as reported by the original paper. This will make the analysis accurate and help to improve the quality of the evidence to support the claim.
  3. Provide the experimental results for more baselines (even if it is limited to memory-based approaches).
- It is highly recommended to start investigating how the recent SoTA are observed and related to the interests of the paper. There are several recent approaches noted before such as CaW, NAT, GraphMixer, DyG2Vec, Dygformer, Todyformer.
- The first contribution of the paper needs to be expanded with respect to the number of appraoches to cover a wider spectrum of approaches to be considered as a survey. Otherwise, claim of the contribution in the introduction can be dropped and left to be a typical related work. To highlight this, it is important to mention that almost 6 pages of the paper are dedicated to the survey while still recent SoTA are left behind.
- There is a lack of motivation behind why studying the tuncation gap in CTDG is important while the real problems require processing of ever-growing dynamic graphs using limited computational resources. It reminds of the gap between the Gradient Descent and the Stochastic Gradient Descent. Furthermore, the truncation might be regarded as a welcoming mechanism that would impose some regularization on learning.
- Elaboration on this on page 6 is needed "Finally, Zhou et al. (2023) propose efficient ways to distribute the training of the above models over multiple
GPUs, but its proposed methods do not affect the truncation problem"
- This statement does not seem accurate: "Because this flaw is tightly linked to the use of backpropagation, and due to the success of backpropagation in other settings and the ease of implementation in current deep learning frameworks, it has been accepted as an unavoidable shortcoming". BPTT in RNN has been a challenge in the DL community while recent years, Transformer-based (or recent variants of the state-space models) appraoch have shown to be able to overcome some of the shortcomings.
- There is a lack of motivation on how the synthetic task helps understand the truncation gap caused by the batching strategies noted in Sec. 4; "In order to investigate the effect of limiting the backpropragation to a single batch in GRNNs on their ability to learn longer term dependencies we propose a novel edge regression synthetic task". This is a reference to the point that the gap between the claim and evidence start to increase where the setup is designed for a synthetic task to highlight the shortcoming of RNN to learn using BPTT. It should be motivated how this directly related to CTDG as CTDG are not sequential data where RNN are proppsed to represent.
- It should be emphasized how the claim on the truncation gap is based on the presumptions on memory-based methods such as chronological training. If the presumptions are voided, does the truncation gap issue is no-longer valid. Elaboration on this point is needed.
- Fig 4.b seems problematic where (4+10)/2=7 and should go to (h^2)_0 according to the earlier eq.  while is gone to (h^1)_0.
- The notatation (h^(s_k)_(k))_M seems slighly ambigious particuarly when it comes to the subscript k and k-1 as it is not clear what it aims to show. Explicit delineation is very helpful to further distangle the notation.
- the synthetic data generation does not completely mimick the distribution of dynamic graph approached in CTDGs. The assumption that not only the order of edges but also the time interval between them matters. A node may have an interation at the edge sequence order of 0 and 10 but the interval between the two might range from 1 second all the way to one year. Consequently, the time scale is neglicted in this experimental setup and needs to be aligned with respect to the common experimental specification in CTDGs.
- It is too late to clarify on the fact that the aim of the paper is not to challenge SoTA at the end of page 8; "These choices are made to
simplify the experimental setup since our goal is not to challenge the state of the art results for dynamic graphs.". The narration of the paper at the beginning has been towards the delineating fact that the SoTA for CTDG inherent the truncation gap and hence suffers from the lack of performance. While the claim is high the supporting evidence stemmed from the over-simplified experimental setup by lifting the time scale dyancmis in the empricial studies seems to harm closing the gap between the evidence and claims to be fuuly supported with accurate and convincing claims. As noted before, either the claim needs to be narrowed down to a limited type of CTDGs or the experimental setup upgraded to cover similar setups studied by SoTA hence the evidence would be accurate and convincing.
- further detailed information on the dynamic graph generation is required to clear out with more acucrate details "For our experiments we consider dynamic graphs consisting of 1000 edges and using a total of 100 nodes, which we refer to as one epoch. We generate a new dynamic graph per epoch.". What is the generation process? What is the nature of the nodes and edges?
- It is not clear how the training procedure on the real datasets for the F-BPTT is achieved with mini-batching of size 200 where it was expected to have the full propagation across the entire training set rather than a sum of batches of edges. Under this scenario, F-BPTT seems similar to T-BPTT due to the batching mechanism. Batching is the source of truncation. It is vague that through mini-baching how F-BPTT is different from T-BPTT and how the gradient truncation is done in the latter with respect to the non-truncated one in the former.
- Additionally, there is a significant missing details on how the experiments could be managed to handle computationally datasets like MOOC where there are ~7K nodes and ~400K edges.
- There is a significant gap between the results for the truncated case with the baseline method TGN for all 3 benchmark datasets let alone the results of the recent SoTA. This is another reason the supprting evidence has further been fallen behind the claim of the paper and hence deteriorating closing the gap. It is highly recommended to push the results of the truncated case all the way up to the comperative level with the baseline and then demonstrate and analyze what the truncation gap is. Since the experimental result is too low, the empirical evidence fall behind being able to support the claim in the real world setup.
- under section 6, there is a contrary statement to the part pointed the intention is not to challenge SoTA "While we argued that recurrent neural network architectures are well-suited to handle tasks on dynamic graph data, we have shown that the truncation present in state-of-the-art approaches can lead to the failure of learning tasks requiring larger temporal dependencies"
- Furthermore, the paper seems not accurate on statements such as "More specifically, recent work (Tallec & Ollivier, 2017; Mujika et al., 2018; Benzing et al., 2019) describe unbiased stochastic low-rank approximations to the true Jacobians in recurrent neural networks on sequential data,
which make online learning feasible" where references from 2017-2019 are regarded as recent work.
- an study on the memory and space complexity tradeoff between the two truncation modes would be insightfull.

**Strengths And Weaknesses:**

## Strenghts
- The fundumental appraoch to study the shortcomings of baseline methods in dynamic graphs is very important. Diving into the underlying mechanisms behind the learning dynamics through back-propagation is very insightful.
- Quantifying the learning deficiency of BPTT and measuring the gap between the optimal case with the sub-optimal case is persuaded in the paper.
- The paper aims to open up the details behind the common practices for the batching mechanism in CTDG. This has been very less visited by the research community and it will bring good value for the future research works. Additionally, it is very critical to highlight and distangle the nature of dynamic interatctions in CTDG and why over-simplification of dynamic graphs to sequential data is overdone and very detrimental.

## Weakness
- The surveyed papers are out-dated with respect to the first indicated controbution of the paper. There are recent SoTA such as CaW [1], NAT [2], GraphMixer [3], DyG2Vec [4], Dygformer [5], Todyformer [6] that needs to be cited, studied, and analyzed according to the ascpect the current one are investigated. At the current stage, the claim on the contribution is insufficient with respect to the span of papers surveyed.
- There is an unaddressed gap between the claim and the supporting evidence. The claim is the truncation which is not studied before in CTDGs, limits the learning resulting into lower performance. However, down the road, both the survey, and the formulation are limited to a specifica category of methods limited to memory-based and RNN-based MPNNs for CTDGs while it is not clear what the other categories such as non-memory-based (i.e. Transformer-based) models are doing according to the truncation gap, whether it exists or not, and how it is quantitatively.
- In addition to the divergence on the survey and the formulation and theoretical investigation, the emprical supporting evidence are obtained from a simplified experimental setup on both the synthetic and real datasets where this further increases the gap between the claim and the evidence by straying from the experimental setup of SoTA.
- There is a lack of emprical evidence to prove the claim is properly supported. The experimental evaluation is limited to simple real datasets, to an early baseline (from 2020), there are simplifications such as dropping the temporal information of interactions that might harm the significance of the analysis.
- Due to the limitation to memory-based (RNN) approaches like TGN there are assumption all over the paper that are not accurate to generalize to all the methods for CTDGs. For instane, memory-based methods would require chronological and sequential training while recent Transformer-based approaches like DyG2Vec does not require that.

[1] Yanbang Wang et al. Inductive representation learning in temporal networks via causal anonymous walks. In Proc. Int. Conf. on Learning Representations, 2021.

[2] Yuhong Luo and Pan Li. Neighborhood-aware scalable temporal network representation learning. In The First Learning on Graphs Conference, 2022.

[3] Weilin Cong, et al. Do we really need complicated model architectures for temporal networks? ICLR, 2023.

[4] Mohammad Ali Alomrani, et al. Dyg2vec: Representation
learning for dynamic graphs with self-supervision. TMLR 2022.

[5] Le Yu, et al. Towards better dynamic graph learning: New architecture and unified library. NeurIPS, 2023.

[6] Mahdi Biparva, et al. TodyFormer: Towards holistic dynamic graph transformers with structure-aware tokenization. TMLR, 2024.

---

> ### Author Response · Authors · 2024-10-31
> **Response to Reviewer**
>
> We thank the reviewer for their thorough review of our work. We address the reviewer’s main comments below.
>
> **Comparison to other approaches for dynamic graphs**
>
> Our goal was not to provide an extensive review of all types of approaches for dynamic graphs because the truncation gap problem is indeed exclusive to a specific type of memory based recurrent architecture which we call GRNN. Since this type of memory component is present in many popular architectures, such as TGN, we hope our work can help bring awareness to this problem and suggest a possible path to improve these models in future work.
>
> We have nonetheless adapted the title, and reformulated the Background and Related Work sections of our work to provide some additional context on alternative architectures explaining why they are not subject to the truncation issue and discussing some of their advantages and disadvantages compared to GRNNs. We hope that this frames our contribution more clearly.
>
> **Batching strategy and full backpropagation**
>
> For full backpropagation, the entire sequence of events is processed in batches of 200 using the same batching strategy as TGN. At the end of a full training epoch, we backpropagate starting from the total training loss through the entire sequence of updates. This is only feasible because of the relatively small size of our datasets and the chosen hidden state size.
>
> In contrast, for truncated backpropagation, gradients are computed for each batch independently (how GRNN models are trained in practice). Backpropagation does not cross batch boundaries and therefore simply computes gradients with respect to a single batch of model updates. This ignores the fact that the hidden states that serve as input to each batch are themselves functions of the model parameters. That is, despite the fact that these states came about by potentially multiple updates to the node states using the model before the start of the batch, their dependence on the model parameters is ignored when backpropagating, leading to incorrect gradients.
>
> While we concede that the batching strategy we use introduces an approximation in that it might ignore certain updates within a batch, it is however a wholly distinct concern to truncation of backpropagation. Furthermore, we point out that both full-backpropagation and truncated-backpropagation use the same batching strategy.  We speculate that, if anything, the approximation introduced by TGN-style batching might be more detrimental in the full-backpropagation setting, as approximations over multiple batches are compounded when computing the gradients.
>
> **Experimental Setup and its Representativeness**
>
> The main goal in our paper is comparison of truncated backpropagation in GRNNs (therefore computing incorrect gradients) with non-truncated (computing correct gradients) in order to show that we are indeed losing performance by using the former. Unfortunately, full BPTT is not a viable solution to the truncation gap because it doesn’t scale to large graphs. The choice of datasets and model simplifications are simply a way to enable us to make this comparison in a real world setting. While not ideal and perfectly representative, we believe that the truncation gap is orthogonal to the compromises made in experimental setup.
>
> Furthermore, regarding our experimental setup, we follow the evaluation proposed in [1] rather than the more common AP or ROC AUC evaluations which makes direct comparisons with other works using the latter difficult. As reported in [1], we found AP evaluation setups with negative sampling to be too easy on the benchmark datasets with little sensitivity of the metrics to model improvements (since APs and AUCs are already so close to 1 in these datasets). In particular we achieved similar or better performance to TGN using our GRNN implementation, even with model simplifications such as no neighborhood embedding module and no time embeddings.
> Unfortunately, [1] focuses on larger datasets which we cannot use full backpropagation with but the results we obtained on the one dataset we have in common with [1] (wikipedia) already show a significant improvement for our simplified architecture over those reported for TGN. We hope that this helps dissipate some of the concerns that our setup is not representative.
>
> [1] Shenyang Huang, et al. Temporal Graph Benchmark for Machine Learning on Temporal Graphs. In NeurIPS 2023 Track on Datasets and Benchmarks
>
> **How we handle large datasets like MOOC**
>
> The main challenge with full-backpropagation is that it requires storing the full sequence of node hidden states computed during the forward pass. For MOOC, this would mean 7k node states at the end of each of the ~300k/200 = 1.5k training batches that comprise one epoch. Given that each node state consists of 64 float32 values, this requires ~2.5GB of GPU VRAM to store.

---

> > ### Comment · Reviewer_i7NF · 2024-11-14
> >
> > Thanks for the feedback on how the comments are addressed. Some of the comments are not addressed neither in the feedback nor the revised version of the paper. Below you can find the comments that seem not to be handled at this version of the draft of the paper yet. Please review and revise the paper as needed to consider the following comments.
> >
> > - There is a lack of motivation on how the synthetic task helps understand the truncation gap caused by the batching strategies noted in Sec. 4; "In order to investigate the effect of limiting the backpropragation to a single batch in GRNNs on their ability to learn longer term dependencies we propose a novel edge regression synthetic task". This is a reference to the point that the gap between the claim and evidence start to increase where the setup is designed for a synthetic task to highlight the shortcoming of RNN to learn using BPTT. It should be motivated how this directly related to CTDG as CTDG are not sequential data where RNN are proppsed to represent.
> > - It should be emphasized how the claim on the truncation gap is based on the presumptions on memory-based methods such as chronological training. If the presumptions are voided, does the truncation gap issue is no-longer valid. Elaboration on this point is needed.
> > - The notation (h^(s_k)_(k))_M seems slightly ambiguous particularly when it comes to the subscript k and k-1 as it is not clear what it aims to show. Explicit delineation is very helpful to further distangle the notation.
> > - the synthetic data generation does not completely mimic the distribution of dynamic graph approached in CTDGs. The assumption that not only the order of edges but also the time interval between them matters. A node may have an interaction at the edge sequence order of 0 and 10 but the interval between the two might range from 1 second all the way to one year. Consequently, the time scale is neglected in this experimental setup and needs to be aligned with respect to the common experimental specification in CTDGs.
> > - further detailed information on the dynamic graph generation is required to clear out with more acucrate details "For our experiments we consider dynamic graphs consisting of 1000 edges and using a total of 100 nodes, which we refer to as one epoch. We generate a new dynamic graph per epoch.". What is the generation process? What is the nature of the nodes and edges?
> > - Furthermore, the paper seems not accurate on statements such as "More specifically, recent work (Tallec & Ollivier, 2017; Mujika et al., 2018; Benzing et al., 2019) describe unbiased stochastic low-rank approximations to the true Jacobians in recurrent neural networks on sequential data, which make online learning feasible" where references from 2017-2019 are regarded as recent work.
> > - an study on the memory and space complexity tradeoff between the two truncation modes would be insightful.

---

> > > ### Author Response · Authors · 2024-11-15
> > > **Response to reviewer i7NF**
> > >
> > > We thank the reviewer for the questions and comments. Please find below our answers to each point:
> > >
> > > 1. Regarding the lack of motivation on using the synthetic task:
> > >
> > > We kindly ask the reviewer to elaborate this question a bit more. The synthetic task is not sequence-based, but is in fact a dynamic graph with interacting nodes over time.
> > >
> > > 2. Regarding the assumptions of memory-based methods:
> > >
> > > To train memory-based methods, there is no way around training on chronologically ordered data (a next hidden state depends on the previous state). This is not an assumption that can be altered, but it is part of the memory-based method. If one wishes to learn on such graph data in different ways, one necessarily will move away from memory-based methods and use e.g. transformers. We discuss those alternatives in the first paragraphs of section 2, and state there that the truncation gap does not apply:
> > >
> > > “While, by construction, this limits the extent of past information that the model can leverage, these methods don't suffer from the same issue as RNN based approaches where the computational graph that is back-propagated through is a truncated version of the computational graph used to arrive at a forward prediction.”
> > >
> > > One choice, though, is the training using backpropagation-through-time, and we discuss that this choice is tightly linked to the truncation gap issue:
> > >
> > > Final paragraph of section 2.2: “...this flaw is tightly linked to the use of backpropagation…”.
> > >
> > > We therefore discuss in section 5 which methods could be used instead of backpropagation, to avoid the truncation gap.
> > >
> > > 3. On the notation:
> > >
> > > We have now added the following sentence in section 3.1, to clarify the notation:
> > >
> > > Here, $\left(\mathbf{h}^{(s_k)}_{k-1}\right)_M$ denotes the $M$th element of the internal state for node $s_k$ (superscript) at step $k-1$ (subscript).
> > >
> > > 4. On the synthetic task and time intervals:
> > >
> > > We appreciate the comment by the reviewer. We agree that the interval is typically important in CTDGs. The key message we want to make is that the truncation gaps limits learning dependencies over many hops in the past (which is typically related to long temporal dependencies), as we wrote in the introduction:
> > >
> > > “...only temporal dependencies spanning a single hop in the graph can be learned accurately…”
> > >
> > > We agree that there may be quite some variability in the relation of a hop in the graph and the time interval and we should not use 'temporal dependencies' interchangeably with 'dependencies on hops'. For this reason, we have rephrased the use of ‘temporal dependencies’ throughout the manuscript to “dependencies on multiple hops” where this distinction was not clear.
> > >
> > > As such, it should now be clear that adding intervals to the synthetic task would not make a difference regarding our claims, because we are investigating dependencies on hops (regardless of intervals).
> > >
> > > 5. On the graph generation:
> > >
> > > Thank you for pointing this out. The generation process was actually described in section 3.1, but we have now moved the above sentence right below this description of the generation process to avoid confusion.
> > >
> > > 6. We have removed the mention of ‘recent work’.
> > >
> > > 7. On the memory complexity:
> > >
> > > We would kindly ask the reviewer to clarify this point. Do you mean between truncated and full backpropagation-through-time? In both cases, the complexities at inference are identical. For training, the memory/time complexity of backpropagation-through-time grows linearly with the number of events (so truncating to a single event vs using a sequence of length 100 will differ by a factor of 100). As this are standard properties of backpropagation-through-time, and since these are not the focus of our work, we have not included this in the manuscript.
> > >
> > > We hope that our comments clarified the reviewer's concerns, and we look forward to any further feedback the reviewer may have.

---

### Review · Reviewer_5to8 · 2024-10-27

**Summary Of Contributions:**

The paper examines the effect of using truncated backpropagation through time (TBTT) on the learning of continuous time-dynamic graphs (CTDGs). In the first section of the paper, the authors examine a set eight papers that fit the more general GRNN-based framework that use TBTT to batch and learn parameters. In the second section of the paper, the authors define a synthetic task and measure the performance of FBTT versus TBTT on a set of benchmark datasets.

**Audience:**

Yes

**Claims And Evidence:**

No

**Requested Changes:**

I have read the reviews of the other reviewers as well. In addition to the Weaknesses I outlined above, I agree with reviewer i7NF's assessment that the authors should include the other more recent architectures and also specify in more concrete terms if the truncation gap extends to non-GRNN based models.

**Strengths And Weaknesses:**

**Strengths:**
1. The paper tackles an interesting research problem which can be useful to both machine-learning researchers and practitioners
2. The writing, figure, and notation are all clear and accessible even for non-experts
3. The paper does manage to highlight the severity of truncation in dynamic graphs through Figure 1


**Weaknesses:**

Although the paper tackles a very interesting problem, I interpret the current version of the paper as (a) a brief overview of eight GRNN methods used in the literature, and (b) experiments on a synthetic task on three benchmark datasets. Unfortunately, it does not answer a lot of questions that I would look for in a work of this nature. More specifically:
1. A comprehensive taxonomy chart for the surveyed architectures is not available
2. The surveyed architectures are all very interesting, but as a practitioner I would want to know what the strengths of each architecture are (particularly if the survey is to be a major contribution of this work). Section 3.1. throws some light on how some of the surveyed works use different kinds of batching strategies but none of these are touched upon in the experiments section.
3. As mentioned in the beginning of the Weaknesses section, I currently interpret the experiments section as the performance of FBTT v/s TBTT for the Reddit, Wikipedia, and MOOC datasets. However, these experiments, although certainly useful, do not provide a clear picture of the truncation gap (at least in my mind). For instance, the authors mention that they use the batching strategy from Rossi et al.'s paper which is only one of the many batching strategies they talk about in Section 3.1. It is also not clear why the truncation gap is smaller for Wikipedia than the other two datasets. A comparison between the surveyed methods in this section would be necessary to not only better understand how the truncation gap works but also to assess the severity of the truncation gap in these methods (or if it is even significant).

---

> ### Author Response · Authors · 2024-10-31
> **Response to Reviewer**
>
> We thank the reviewer for their thoughtful review of our work. Below we address the main concerns raised by the reviewer.
>
> **Survey and Clarification of contribution**
>
> Our main goal was to bring attention to an overlooked issue with an important class of models (GRNNs) for dynamic graphs and show that it can have a real-world impact.
> With this we hope to suggest a meaningful direction for improving these models in future work.
> We have rewritten the Introduction, Background and Related Work sections of our paper to focus on this contribution. More concretely, we have clarified our aim in the Introduction, removed the ‘survey-like’ Related Work section as it was distracting from the main message, and only included necessary parts of it in the Background section. We believe that these changes clarify the scope of our work.
>
> **Experimental Setup and Results**
>
> Regarding batching strategies, we agree with the reviewer that it would be interesting to compare their impact on the truncation gap. Our main goal with this section however, was to highlight that all of these strategies lead to the same issue with truncation. The first batching strategy we analyze is impractical as it needs to create a different computational graph for each batch. TGN-like batching is the simplest and most efficient for this purpose, allowing us to jit-compile the entire scan through the graph and train these models for a large number of epochs. While we expect that JODIE-like t-batching could improve performance since it removes the small approximation made with TGN-style batches, it would also be significantly slower and require more GPU memory for full-backpropagation. Furthermore, we speculate that, if anything, the approximation introduced by TGN-style batching might be more detrimental in the full-backpropagation setting, as its approximations over multiple batches are compounded when computing the gradients.
>
> Regarding the results on the Wikipedia dataset, we note that there is still a gap when considering the recall@10 metric even if it is smaller than other datasets. Furthermore, full backpropagation is able to achieve a considerably lower training loss but unfortunately this does not translate to a significant gap in test performance indicating that it is easier to overfit in this dataset. Given the black-box nature of these models and the datasets it is hard to speculate on a reason why this might be so since we do not know the ground-truth in terms of temporal dependencies. It is plausible that on Wikipedia, longer temporal information is not as relevant for the link prediction task, which in turn would result in a small or no truncation gap.

---

### Author Response · Authors · 2024-10-31
**Author Rebuttal**

We thank all reviewers for their time and insightful comments on our paper.

We are pleased that the reviewers recognized our work as addressing a relevant research problem that warrants the attention of the community and found our paper to be well presented and insightful.

Our primary goal was not to provide an exhaustive review and experimental comparison of different types of approaches for temporal graphs. Instead, we aimed to highlight an issue that affects a specific type of memory based recurrent architecture, which we term GRNN, that can impact its practical performance.
Given that this type of memory component is present in many popular models, such as TGN, we hope our work can raise awareness of this issue and suggest a potential direction for improving these models in future work.
To clarify this focus, we have revised the Background and Related Work sections, as several reviewers questioned whether the truncation gap also affects non-recurrent architectures.
We have also included a brief discussion of alternative approaches, noting that while these avoid the truncation issue of GRNNs, it is because they are limiting by design the context that is considered for each prediction.

Furthermore, in order to make the message clear and set the right expectations from the start we have:
- Adjusted the title of the paper to reflect the focus on recurrent architectures
- Clearly stated in the introduction that the main goal of the paper is to investigate the impact of truncating backpropagation in GRNN architectures
- Removed the claim of a survey as a main contribution of the paper
- Fixed the error pointed out by reviewer `i7NF` in (what is now) Figure 3

We hope that these changes address the main concerns raised by the reviewers. Specific comments have been addressed in line with individual reviewer feedback (see replies to each reviewer).
We believe these changes significantly improve the clarity and focus of our work, and we look forward to any further feedback the reviewers may have.

---

### Decision · Action_Editor_LwR7 · 2024-11-22

**Recommendation:** Accept as is

**Comment:**

This decision is a little tricky, since the reviewers span all 4 possible options (Accept, Leaning Accept, Leaning Reject, Reject). It is worth noting that, of the 3 reviewers I invited to review this paper, 2/3 recommended accept. The 4th reviewer (who recommended Leaning Reject) volunteered to review.

From reading the reviews of the reviewers' that recommend reject, the weaknesses identified are focused on: 1) not being state-of-the-art, 2) not providing a wide enough investigation of all the architectural/hyper-parameter properties that affect the results. While I believe these are limitations of this work, they are not sufficient to reject the paper from the TMLR acceptance criteria.

For the above reason, I recommend accepting the paper. The authors updated their manuscript to meet the comments of the reviewers, and I do not think additional minor editing would provide much change. Therefore, I am recommending accepting as is.

**Audience:**

All reviewers agreed that there was a sufficient audience in TMLR's community that would be interested in the results of this paper.

**Claims And Evidence:**

While 3 of the 4 reviewers initially felt that there was not sufficient evidence supporting the authors' claims, the authors provided significant responses to the reviewers. They addressed many of the concerns and clarified points that had not be clear to reviewers initially. Therefore, I believe that the revised manuscript is supported by sufficient evidence.